

# Prognostic value of apparent diffusion coefficient in neuroendocrine carcinomas of the uterine cervix

Jian Chen[1], Ning Ma[2], Mingyao Sun[3], Li Chen[1], Qimin Yao[4], XingFa Chen[2], Cuibo Lin[1], Yongwei Lu[1], Yingtao Lin[5], Liang Lin[1], Xuexiong Fan[6], Yiyu Chen[7], Jingjing Wu[1] and Haixin He[1]

[1] Department of Gynecology, Clinical Oncology School of Fujian Medical University, Fujian Cancer Hospital, Fuzhou, Fujian, China

[2] Department of Radiology, Clinical Oncology School of Fujian Medical University, Fujian Cancer Hospital, Fuzhou, Fujian, China

[3] Department of Clinical Nutrition, Fujian Provincial Hospital, Fuzhou, Fujian, China

[4] College of Finance, Fujian Jiangxia University, Fuzhou, Fujian, China

[5] Department of Drug Clinical Trial Institution, Clinical Oncology School of Fujian Medical University, Fujian Cancer Hospital, Fuzhou, Fujian, China

[6] Department of Medical Record, Clinical Oncology School of Fujian Medical University, Fujian Cancer Hospital, Fuzhou, Fujian, China

[7] Department of Pathology, Clinical Oncology School of Fujian Medical University, Fujian Cancer Hospital, Fuzhou, Fujian, China

Corresponding author
Haixin He, 63804657@qq.com

## ABSTRACT

**Objectives**. This research was designed to examine the associations between the apparent diffusion coefficient (ADC) values and clinicopathological parameters, and to explore the prognostic value of ADC values in predicting the International Federation of Gynecology and Obstetrics (FIGO) stage and outcome of patients suffering from neuroendocrine carcinomas of the uterine cervix (NECCs).

**Methods**. This retrospective study included 83 patients with NECCs, who had undergone pre-treatment magnetic resonance imaging (MRI) between November 2002 and June 2019. The median follow-up period was 50.7 months. Regions of interest (ROIs) were drawn manually by two radiologists. ADC values in the lesions were calculated using the Functool software. These values were compared between different clinicopathological parameters groups. The Kaplan–Meier approach was adopted to forecast survival rates. Prognostic factors were decided by the Cox regression method.

**Results**. In the cohort of 83 patients, nine, 42, 23, and nine patients were in stage I, II, III, and IV, respectively. $ADC_{mean}$, $ADC_{max}$, and $ADC_{min}$ were greatly lower in stage IIB–IVB than in stage I–IIA tumours, as well as in tumours measuring $\geq 4$ cm than in those $< 4$ cm. $ADC_{mean}$, FIGO stage, and age at dianosis were independent prognostic variables for the 5-year overall survival (OS). $ADC_{min}$, FIGO stage, age at diagnosis and para-aortic lymph node metastasis were independent prognostic variables for the 5-year progression-free survival (PFS) in multivariate analysis. For surgically treated patients ($n = 45$), $ADC_{max}$ was an independent prognostic parameter for both 5-year OS and 5-year PFS.

**Conclusions**. $ADC_{mean}$, $ADC_{min}$, and $ADC_{max}$ are independent prognostic factors for NECCs. ADC analysis could be useful in predicting the survival outcomes in patients with NECCs.

## INTRODUCTION

Neuroendocrine carcinomas of the cervix (NECCs), an infrequent but highly invasive form of cervical cancer, represent less than 5% of cervical carcinomas (*Gardner, Reidy-Lagunes & Gehrig, 2011*; *Li et al., 2020*; *Lin et al., 2020*; *Satoh et al., 2014*). According to the World Health Organization classification, NECCs are divided into small cell neuroendocrine carcinomas (SCNECs) and large cell neuroendocrine carcinomas (LCNECs) (*Satoh et al., 2014*). Owe to the high incidence of early lymph node involvement and distant metastasis, the outcome of patients with NECCs is worse than those with other subtypes of cervical cancer. Radical surgery and chemoradiation are recommended as the primary treatment for patients with early- and advanced-stage disease, respectively (*Bhatla et al., 2019*). Due to the rarity of NECCs, most studies on NECCs are reported in small samples or are case reports (*McCann et al., 2013*; *Yuan et al., 2015*). Therefore, the prognostic parameters and treatment of NECCs are controversial. Advanced FIGO stage, lymph node involvement, large tumor size, older age and lymphovascular invasion have been reported to be associated with poor prognosis (*Chen et al., 2021*; *Gadducci, Carinelli & Aletti, 2017*). However, the described factors play a limited role in predicting the prognosis of NECCs.

MRI is a great tool for diagnosing and staging cervical tumours due to its high soft tissue resolution. On T1- and T2-weighted images (T1WI and T2WI, respectively) as well as contrast-enhanced images, MRI can show both morphologic and signal intensity properties (*Nakamura et al., 2012*). Diffusion-weighted imaging (DWI), a functional imaging technology, quantifies the free movement of water molecules (Brownian molecular movement) through ADC values (*Bruix & Llovet, 2002*; *Fan et al., 2020*). ADC describes the velocity and scope of molecular diffusion movements in various directions (*Liang et al., 2016*). Moreover, ADC values provide useful information about tumour aggressiveness, subtype characterisation, and treatment responses taking into consideration the limiting barriers in tissue compartments (*De Robertis et al., 2018*; *Gu et al., 2019*; *Meyer et al., 2019*; *Perucho et al., 2020*; *Zou et al., 2019*). *Schob et al. (2017a)* found that ADC value were useful in predicting lymphatic metastasis, and proliferative activity in thyroid cancer. In gastric cancer, *Liu et al. (2014)* demonstrated that ADC analysis is helpful to assess the pre-treatment T and N staging. To date, few studies have discussed the application of MRI in NECCs. *Duan et al. (2016)* found that NECCs are characterized by lower ADC values and homogeneous lesion texture on MRI images. However, no studies have reported the utility of ADC values in predicting the outcomes of patients with NECCs. In this study, we examined the associations between maximum, mean, and minimum ADC values ($ADC_{max}$, $ADC_{mean}$ and $ADC_{min}$, respectively) and clinicopathological parameters, as well as the prognostic value of ADC values in predicting the stage and outcome in patients with NECCs, in a retrospective review of 83 patients. We also assessed the accuracy of MRI in the diagnosis of NECCs.

## MATERIALS & METHODS

### Patients and treatment

The research was approved by the Ethics Committee of Fujian Medical University Cancer Hospital (Reference No: K2021-043-01). Between November 2002 and June 2019, the clinicopathological information of 172 patients with pathological confirmed NECCs who received treatment at Fujian Medical University Cancer Hospital, were reviewed. All histologic slides were reviewed by two experienced pathologists to confirm the diagnosis of NECCs. The requirement for informed consent was waived due to the retrospective nature of this study. The following were the standards for inclusion: (1) those with pathological confirmed NECCs, (2) those who underwent pre-treatment abdominal and pelvic MRI in our centre, and (3) those who received treatment in our centre and had complete medical records. The exclusion standards were as shown: (1) presence of other concurrent malignancies, (2) history of cancer, (3) those who refused or discontinued treatment, and (4) lost to follow-up. Finally, 83 patients were recruited in the group.

### MRI imaging

A 1.5T MRI system (GE Signa HDxT) was used. Before the examination, the patients were required to drink enough water to fill their bladder to a moderate level. The MRI scan extended from the renal hilum to the perineum. Routine abdominal and pelvic MRI including the following sequences were acquired as follows: (1) sagittal T2WI: fast spin-echo (FSE) sequence, repetition time (TR)/echo time (TE), 4760/104 ms; matrix size, $320 \times 192$; field of view (FOV), 24 cm; slice thickness/intersection gap, 5/1 mm; (2) axial T2WI: TR/TE, 4320/105 ms; matrix size, $320 \times 192$; FOV, 24 cm; slice thickness/intersection gap, 5/1 mm; (3) axial T1WI: fast FSE sequence; TR/TE, 600/7.7 ms; matrix size, $320 \times 256$; FOV, 48 cm; slice thickness/intersection gap, 7/1 mm. The protocols for axial DWI ($b = 0$, 800 s/mm$^2$) were as follows: TR, 4225 ms, TE, minimum time; matrix size, $128 \times 128$; FOV, 38 cm; slice thickness/intersection gap, 7/1 mm.

### Imaging analysis

All images were retrieved from the local picture archiving and communication systems. Two radiologists analysed the MR images in consensus (N.M. and X.F.C, with 8 and 12 years of experience in gynaecologic imaging, respectively). The two radiologists agreed on the criteria for determining the tumour size, vaginal extension, parametrial extension, and lymph node metastasis according to the FIGO 2018 staging and the guidance of the European Society of Urogenital Radiology (*Balleyguier et al., 2011*; *Bhatla et al., 2018*). They were blinded to the patients' information. The ADC map was constructed automatically using the Functool software on the Advantage Workstation(AW 4.2 version, GE, US. https://www.medicalexpo.com/product-manufacturer/ge-mri-system-15892-438.html). Regions of interest (ROIs) were manually drawn along the margin of the lesions showing maximal tumour size on axial DWI images. The ROI did not include parts of the tumour that were cystic, necrotic, or haemorrhagic. ADC values in the lesions were calculated using the software (Fig. 1).

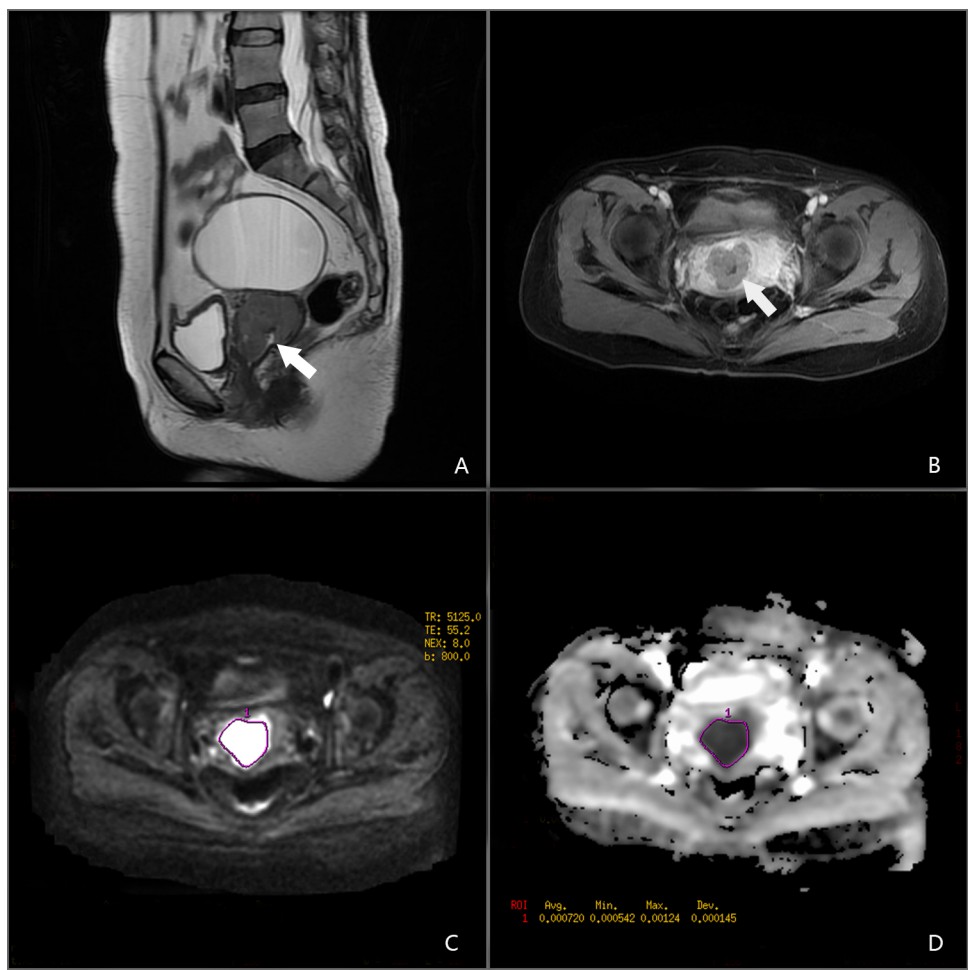

**Figure 1** **A 45-year-old female patient with NECC.** (A) Sagittal T2-weighted image. (B) Axial T1-enhanced image. (C) DWI in $b = 800$. (D) ADC map.

## Treatment

Since this was a retrospective study, therefore, the treatment plans of most cases were largely dependent on FIGO 2008 guidelines. Surgery was the primary treatment for patients in the early stage (FIGO stage I–IIA), and chemoradiation was applied for patients in the advanced stage (FIGO stage III–IV). For patients with IIB stage, whether to undergo surgery after neoadjuvant therapy or received chemoradiation treatment depends on the doctor's judgment. According to the postoperative pathology report, adjuvant therapy was performed if there were risk factors such as lymph node metastases, positive margin, deep stromal invasion, lymphovascular invasion, parametrial invasion, and perineural invasion. Finally, 45 patients received surgery and 38 patients received chemoradiation or only chemotherapy. Of the 80 patients who received chemotherapy, 38 received etoposide and cisplatin/carboplatin (EP regimen), and 34 received paclitaxel and cisplatin/carboplatin (TP regimen). The other eight patients received bleomycin, ifosfamide, and cisplatin (one

case); paclitaxel, etoposide, and cisplatin (three cases); docetaxel and platinum (two cases); and TP and EP successively (two cases).

## Statistical analysis

ADC values are shown as mean $\pm$ SD. The normality of all the data was tested by applying the Kolmogorov–Smirnov test. The Student's $t$-test or the Mann–Whitney U test was adopted to compare the ADC values among different tumour groups. Applying the maximum Youden's index, the ROC curve was used to estimate the parameter cutoff values. The Kaplan–Meier and Cox regression methods were adopted to calculate prognostic factors for OS and PFS. The multivariate analysis further investigated prognostic parameters ($p$ values <0.1) from the univariate analysis. $P$ value <0.05 was regarded statistically significant. The SPSS version 24.0 statistical software (SPSS Inc. Chicago, IL, USA, http://www.spss.com) was employed to perform all statistical analyses.

# RESULTS

## Clinicopathologic characteristics of the patients

The clinicopathological characteristics of the patients are listed in Table 1. Eighty-three patients were enrolled; their ages ranged from 25 to 78 years, and the average age was 49.2 years. The average size of the cervical tumour was 4.7 cm. On the basis of the FIGO 2018 staging system, nine, 42, 23, and nine patients were in stage I, II, III, and IV, respectively. There were 80 and three cases of SCNEC and LCNEC, respectively. Pure histology was documented in 68.7% (57/83) of the patients. The mixed histology patterns included adenocarcinoma (16/83 patients; 19.3%), squamous cell carcinoma (7/83 patients; 8.4%), and adenosquamous carcinoma (3/83 patients; 3.6%). In the cohort of 83 patients, the mean values of $ADC_{max}$, $ADC_{mean}$, and $ADC_{min}$ were 0.969, 0.750 and 0.632 ($\times 10^{-3}$ mm$^2$/s), respectively.

## Associations between ADC values and clinicopathological parameters

The results of the Mann–Whitney $U$ test and Student's $t$-test for the comparison between ADC values and clinicopathological features are presented in Table 2. The $ADC_{mean}$ of the primary tumour had a high correlation with the FIGO stage (I-IIA $vs.$ IIB-IVB, 0.880 $\pm$ 0.327 $vs.$ 0.655 $\pm$ 0.226, $p = 0.001$), tumour size (<4 cm $vs.$ >4 cm, 0.882 $\pm$ 0.305 $vs.$ 0.696 $\pm$ 0.274, $p = 0.002$), pelvic lymph node metastasis (negative $vs$ positive, 0.788 $\pm$ 0.332 $vs.$ 0.672 $\pm$ 0.172, $p = 0.04$), and depth of stromal invasion (inner third $vs$ middle to outer third, 0.975 $\pm$ 0.378 $vs.$ 0.748 $\pm$ 0.264, $p = 0.04$). The $ADC_{max}$ of the pre-treatment tumour was greatly related to tumour size (<4 cm $vs.$ >4 cm, 1.069 $\pm$ 0.269 $vs.$ 0.928 $\pm$ 0.282, $p = 0.022$), FIGO stage (I-IIA $vs.$ IIB-IVB, 1.068 $\pm$ 0.299 $vs.$ 0.896 $\pm$ 0.252, $p = 0.01$), and depth of stromal invasion (inner third $vs$ middle to outer third, 1.162 $\pm$ 0.352 $vs.$ 0.951 $\pm$ 0.264, $p = 0.023$). The $ADC_{min}$ was significantly associated with tumour size (<4 cm $vs.$ >4 cm, 0.751 $\pm$ 0.335 $vs.$ 0.583 $\pm$ 0.262, $p = 0.007$), FIGO stage (I-IIA $vs.$ IIB-IVB, 0.761 $\pm$ 0.345 $vs.$ 0.538 $\pm$ 0.206, $p = 0.001$), and pelvic lymph node metastasis (negative $vs$ positive, 0.678 $\pm$ 0.329 $vs.$ 0.534 $\pm$ 0.165, $p = 0.009$).

**Table 1  Patient characteristics ($N = 83$).**

| Variables | Number(%) |
|---|---|
| Hystological type | |
| Small cell neuroendocrine carcinoma | 80(96.4%) |
| Large cell neuroendocrine carcinoma | 3(3.6%) |
| Histological homology | |
| Pure | 57(68.7%) |
| Mix with squamous | 7(8.4%) |
| Mix with adenocarcinoma | 16(19.3%) |
| Mix with adenosquamous carcinoma | 3(3.6%) |
| FIGO stage(2018) | |
| I | |
| IB2 | 6(7.2%) |
| IB3 | 3(3.6%) |
| II | |
| IIA1 | 13(15.7%) |
| IIA2 | 13(15.7%) |
| IIB | 16(19.3%) |
| III | |
| IIIA | 3(3.6%) |
| IIIC1 | 15(18.1%) |
| IIIC2 | 5(6.0%) |
| IV | |
| IVB | 9(10.8%) |
| Lymph node metastasis | |
| Pelvic only | 19(22.9%) |
| Pelvic and para-aortic | 8(9.6%) |
| Negative | 56(67.5%) |
| Primary treatment | |
| Surgery + adjuvant therapy | 12(14.5%) |
| NACT + surgery ± adjuvant therapy | 31(37.3%) |
| Surgery alone | 2(2.4%) |
| CCRT + CT | 30(36.1%) |
| CT alone | 8(9.6%) |
| Chemotherapy regimen | |
| EP | 38(45.8%) |
| TP | 34(41.0%) |
| Other regimens | 8(9.6%) |
| Without chemotherapy | 3(3.6%) |
| Age, years(mean ± SD) | 49.2 ± 10.5 |
| Tumor size,cm(mean ± SD) | 4.7 ± 1.9 |
| $ADC_{mean}$ ($10^{-3}$mm$^2$/s,mean ± SD) | 0.750 ± 0.293 |
| $ADC_{max}$ ($10^{-3}$mm$^2$/s,mean ± SD) | 0.969 ± 0.284 |
| $ADC_{min}$ ($10^{-3}$mm$^2$/s,mean ± SD) | 0.632 ± 0.293 |

**Notes.**
FIGO, International Federation of Gynecology and Obstetrics Adjuvant therapy includes chemotherapy, radiotherapy and concurrent chemoradiation; CT, chemotherapy; CCRT, concurrent chemoradiation; NACT, neoadjuvant chemotherapy.

**Table 2 ADC values and clinicopathological parameters.**

| Variable | N | ADC^mean (10⁻³mm²/s) | P | ADC^max (10⁻³mm²/s) | P | ADC^min (10⁻³mm²/s) | P |
|---|---|---|---|---|---|---|---|
| FIGO stage | | | 0.001* | | 0.010* | | 0.001* |
| I-IIA | 35 | $0.880 \pm 0.327$ | | $1.068 \pm 0.299$ | | $0.761 \pm 0.345$ | |
| IIB-IVB | 48 | $0.655 \pm 0.226$ | | $0.896 \pm 0.252$ | | $0.538 \pm 0.206$ | |
| Age, years | | | 0.630 | | 0.380 | | 0.628 |
| ≦45 | 30 | $0.729 \pm 0.308$ | | $0.932 \pm 0.306$ | | $0.612 \pm 0.303$ | |
| >45 | 53 | $0.762 \pm 0.287$ | | $0.989 \pm 0.272$ | | $0.642 \pm 0.290$ | |
| Tumor size (cm) | | | 0.002* | | 0.022* | | 0.007* |
| <4 | 24 | $0.882 \pm 0.305$ | | $1.069 \pm 0.269$ | | $0.751 \pm 0.335$ | |
| ≧4 | 59 | $0.696 \pm 0.274$ | | $0.928 \pm 0.282$ | | $0.583 \pm 0.262$ | |
| Pelvic LN metastasis | | | 0.040 | | 0.130 | | 0.009 |
| No | 56 | $0.788 \pm 0.332$ | | $1.002 \pm 0.304$ | | $0.678 \pm 0.329$ | |
| Yes | 27 | $0.672 \pm 0.172$ | | $0.901 \pm 0.228$ | | $0.534 \pm 0.165$ | |
| Para-aortic LN metastasis | | | 0.408 | | 0.193 | | 0.346 |
| Negative | 75 | $0.759 \pm 0.304$ | | $0.982 \pm 0.292$ | | $0.642 \pm 0.304$ | |
| Positive | 8 | $0.668 \pm 0.148$ | | $0.844 \pm 0.165$ | | $0.538 \pm 0.131$ | |
| Lymphovascular invasion | | | 0.067 | | 0.064 | | 0.074 |
| Negative | 25 | $0.914 \pm 0.347$ | | $1.108 \pm 0.327$ | | $0.799 \pm 0.345$ | |
| Positive | 20 | $0.735 \pm 0.278$ | | $0.934 \pm 0.273$ | | $0.619 \pm 0.301$ | |
| Depth of stromal invasion | | | 0.040 | | 0.023 | | 0.055 |
| Inner third | 17 | $0.975 \pm 0.378$ | | $1.162 \pm 0.352$ | | $0.852 \pm 0.380$ | |
| Middle to outer third | 28 | $0.748 \pm 0.264$ | | $0.951 \pm 0.264$ | | $0.638 \pm 0.281$ | |
| Histological homology | | | 0.773 | | 0.719 | | 0.407 |
| Pure | 57 | $0.744 \pm 0.288$ | | $0.961 \pm 0.296$ | | $0.614 \pm 0.294$ | |
| Mixed | 26 | $0.764 \pm 0.310$ | | $0.985 \pm 0.263$ | | $0.671 \pm 0.292$ | |

**Notes.**
*P values were dervied applying the Mann-Whitney U test; other P values were dervied applying the Student's t test.

## MRI and pathological staging of NECCs

Among the 45 patients who receiving surgery, 30 patients receiving neoadjuvant therapy were excluded, and finally, 15 patients were included in the analysis to calculate the MRI accuracy. Table 3 shows the agreement between the MRI stage and the pathological stage. The overall accuracy of MRI was only 46% (7/15). Errors were seen in eight patients due to false-negative ($n = 2$) or false-positive ($n = 3$) vaginal invasion, false-negative lymph node metastasis ($n = 1$), or false-positive parametrial invasion ($n = 2$). The accuracy rates of MRI in the diagnosis of uterine corpus invasion, parametrial invasion, vaginal invasion, and lymph node metastasis were 86.7%, 80.0%, 53.3%, and 93.3%, respectively.

## Survival results

The median OS and PFS of the enrolled 83 patients were 42.7 and 38.1 months, and the 5-year OS and PFS rates were 46.3 and 41.4%, respectively. The median follow-up period for all patients was 50.7 months (range: 2–193 months). At the end of follow-up period, cancer recurrence was observed in 47 patients, 41 patients had died. Patients with stage

**Table 3 Comparison of MRI staging and pathological staging in surgically treated patients without neoadjuvant treatment.**

| Parameter | MRI | Pathology | | Sensitivity | Specificity | Accuracy |
|---|---|---|---|---|---|---|
| | | Positive | Negative | | | |
| Uterine corpus invasion | Positive | 0 | 2 | – | 86.7% | 86.7% |
| | Negative | 0 | 13 | | | |
| Parametrial invasion | Positive | 0 | 3 | – | 80.0% | 80.0% |
| | Negative | 0 | 12 | | | |
| Vaginal invasion | Positive | 1 | 7 | 100.0% | 50.0% | 53.3% |
| | Negative | 0 | 7 | | | |
| Lymph node metastasis | Positive | 1 | 0 | 50.0% | 100.0% | 93.3% |
| | Negative | 1 | 13 | | | |

I, II, III, and IV disease had 5-year OS rates of 88.9, 54.6, 35.5, and 0%, respectively. Patients with stage I, II, III, and IV disease had 5-year PFS rates of 77.8, 53.9, 21.9, and 0%, respectively. ROC curve analyses were performed to decide whether ADC values predicted the prognosis of patients diagnosed with NECCs. The optimal $ADC_{mean}$, $ADC_{max}$, and $ADC_{min}$ cutoff values for OS were $0.701 \times 10^{-3}$ mm$^2$/s, $1.041 \times 10^{-3}$ mm$^2$/s, and $0.822 \times 10^{-3}$ mm$^2$/s (AUC: 0.680, 0.717 and 0.614), respectively. The optimal $ADC_{mean}$, $ADC_{max}$, and $ADC_{min}$ cutoff values for PFS were $0.969 \times 10^{-3}$ mm$^2$/ss, $0.997 \times 10^{-3}$ mm$^2$/ss, and $0.922 \times 10^{-3}$ mm$^2$/s (AUC: 0.664, 0.696, and 0.633), respectively (Fig. 2).

## Prognostic factors

Multivariate analyses showed that $ADC_{mean}$ ($\leq 0.7 \times 10^{-3}$ mm$^2$/s vs. $>0.7 \times 10^{-3}$ mm$^2$/s, HR =2.344, CI 95% [1.155–4.756], $p = 0.018$), advanced FIGO stage (HR =2.085, CI 95%[1.351–3.217], $p = 0.001$), and age (>45 vs. $\leq 45$ years, HR =2.651, CI 95% [1.257–5.590], $p = 0.01$) were independent prognostic factors for OS. Besides, $ADC_{min}$ ($\leq 0.68 \times 10^{-3}$ mm$^2$/s vs. $>0.68 \times 10^{-3}$ mm$^2$/s, HR =3.787, CI 95% [1.469–9.765], $p = 0.006$), FIGO stage (HR =1.919, CI 95% [1.221–3.014], $p = 0.005$), age (>45 vs. $\leq 45$ years, HR =2.380, CI 95% [1.222–4.635], $p = 0.011$) , and para-aortic lymph node metastasis (positive vs negative, HR =3.151, CI 95% [1.204–8.248], $p = 0.019$) were significant prognostic parameters for PFS (Table 4). Survival curves for patients with various ADC values, FIGO stage, ages, and para-aortic lymph node statuses were shown in the Figs. 3 and 4. In patients who received surgery, patients with $ADC_{max}$ $>1.032 \times 10^{-3}$ mm$^2$/s had significantly better 5-year OS (92.9% vs. 36.9%, $p = 0.006$) and 5-year PFS (83.7% vs. 30.0%, $p = 0.006$) rates than those with $ADC_{max}$ $\leq 1.032 \times 10^{-3}$ mm$^2$/s. Besides, lymphovascular invasion was another prognostic factor that affected OS (negative vs. positive: 76.1% vs. 38.3%; $p = 0.009$) (Table 5).

## DISCUSSION

To the best of our knowledge, this study was the first to investigate the prognostic utility of ADC values in predicting the outcomes of patients with NECCs. Moreover, the correlation between the ADC values and clinicopathological parameters in neuroendocrine carcinomas

 

Chen et al. (2023), *PeerJ*, DOI 10.7717/peerj.15084

**Table 4  Univariate and multivariate analysis of clinicopathological and treatment parameters for the all series ($n = 83$).**

| Variable | n | Overall survival | | | | | | Progression free survival | | | | | |
|---|---|---|---|---|---|---|---|---|---|---|---|---|---|
| | | Univariate | | | Multivariate | | | Univariate | | | Multivariate | | |
| | | HR | 95%CI | P | HR | 95%CI | P | HR | 95%CI | P | HR | 95%CI | P |
| Hystological type | | 1.537 | 0.369-6.406 | 0.555 | | | | 1.424 | 0.344-5.893 | 0.626 | | | |
| SCNEC | 80 | | | | | | | | | | | | |
| LCNEC | 3 | | | | | | | | | | | | |
| Age, years | | 2.552 | 1.236-5.270 | 0.011 | 2.651 | 1.257-5.590 | 0.01 | 2.117 | 1.106-4.054 | 0.024 | 2.380 | 1.222-4.635 | 0.011 |
| ≤45 | 30 | | | | | | | | | | | | |
| >45 | 53 | | | | | | | | | | | | |
| Tumor size(cm) | | 2.225 | 1.061-6.047 | 0.036 | 1.354 | 0.539-3.403 | 0.519 | 1.639 | 0.811-3.311 | 0.169 | | | |
| <4 | 24 | | | | | | | | | | | | |
| ≥4 | 59 | | | | | | | | | | | | |
| FIGO stage(2018) | | 2.339 | 1.578-3.469 | 0.001 | 2.085 | 1.351-3.217 | 0.001 | 2.423 | 1.634-3.342 | ¡0.001 | 1.919 | 1.221-3.014 | 0.005 |
| I | 9 | | | | | | | | | | | | |
| II | 42 | | | | | | | | | | | | |
| III | 23 | | | | | | | | | | | | |
| IV | 9 | | | | | | | | | | | | |
| Histological homology | | 1.198 | 0.666-2.534 | 0.443 | | | | 1.425 | 0.772-2.629 | 0.357 | | | |
| Pure | 57 | | | | | | | | | | | | |
| Mixed | 26 | | | | | | | | | | | | |
| Pelvic LN metastasis | | 2.557 | 1.477-5.316 | 0.002 | 1.138 | 0.440-2.946 | 0.789 | 3.405 | 1.872-6.191 | ¡0.001 | 1.576 | 0.615-4.036 | 0.343 |
| No | 56 | | | | | | | | | | | | |
| Yes | 27 | | | | | | | | | | | | |
| Para-aortic LN metastasis | | 3.287 | 1.573-8.203 | 0.002 | 1.265 | 0.439-3.649 | 0.663 | 5.241 | 2.405-11.422 | ¡0.001 | 3.151 | 1.204-8.248 | 0.019 |
| No | 75 | | | | | | | | | | | | |
| Yes | 8 | | | | | | | | | | | | |
| Chemotherapy regimen | | 1.079 | 0.717-1.767 | 0.608 | | | | 1.075 | 0.713-1.622 | 0.73 | | | |
| TP | 34 | | | | | | | | | | | | |
| EP | 38 | | | | | | | | | | | | |
| Other regimens | 8 | | | | | | | | | | | | |
| Without chemotherapy | 3 | | | | | | | | | | | | |
| Cycle of chemotherapy | | 0.556 | 0.286-1.084 | 0.085 | 0.546 | 0.278-1.071 | 0.078 | 0.771 | 0.424-1.401 | 0.394 | | | |
| 0-5 | 48 | | | | | | | | | | | | |
| ≥6 | 35 | | | | | | | | | | | | |

Chen et al. (2023), *PeerJ*, DOI 10.7717/peerj.15084

**Table 4** (*continued*)

| Variable | n | Overall survival | | | | | | Progression free survival | | | | | |
|---|---|---|---|---|---|---|---|---|---|---|---|---|---|
| | | Univariate | | | Multivariate | | | Univariate | | | Multivariate | | |
| | | HR | 95%CI | *P* | HR | 95%CI | *P* | HR | 95%CI | *P* | HR | 95%CI | *P* |
| $ADC_{min}(10^{-3}mm^2/s)$ | | 3.250 | 1.269-8.323 | 0.014 | 1.286 | 0.368-4.498 | 0.694 | 3.147 | 1.328-7.458 | 0.009 | 3.787 | 1.469-9.765 | 0.006 |
| <0.680 | 61 | | | | | | | | | | | | |
| ≥0.680 | 22 | | | | | | | | | | | | |
| $ADC_{max}(10^{-3}mm^2/s)$ | | 4.174 | 1.632-10.68 | 0.003 | 1.853 | 0.622-5.526 | 0.268 | 3.047 | 1.418-6.547 | 0.004 | 1.464 | 0.627-3.423 | 0.379 |
| ≤1.032 | 53 | | | | | | | | | | | | |
| >1.032 | 30 | | | | | | | | | | | | |
| $ADCmean(10^{-3}mm^2/s)$ | | 2.646 | 1.339-5.230 | 0.005 | 2.344 | 1.155-4.756 | 0.018 | 2.140 | 1.161-3.944 | 0.015 | 1.243 | 0.572-2.699 | 0.583 |
| ≤0.700 | 42 | | | | | | | | | | | | |
| >0.700 | 41 | | | | | | | | | | | | |

**Notes.**

SCNEC, Small cell neuroendocrine carcinoma; LCNEC, large cell neuroendocrine carcinoma; Adjuvant therapy includes chemotherapy, radiotherapy and concurrent chemoradiation; CT, chemotherapy; CCRT, concurrent chemoradiation; NACT, neoadjuvant chemotherapy; EP, etoposide and cisplatin/carboplatin; TP, paclitaxel and cisplatin/carboplatin.

Chen et al. (2023), *PeerJ*, DOI 10.7717/peerj.15084

**Table 5  Univariate and multivariate analysis of clinicopathological and treatment parameters for surgically treated patients ($n = 45$).**

| Variable | n | Overall survival | | | | | | Progression free survival | | | | | |
|---|---|---|---|---|---|---|---|---|---|---|---|---|---|
| | | Univariate | | | Multivariate | | | Univariate | | | Multivariate | | |
| | | HR | 95%CI | P | HR | 95%CI | P | HR | 95%CI | P | HR | 95%CI | P |
| Hystological type | | 1.194 | 0.157-9.104 | 0.864 | | | | 0.960 | 0.128-7.202 | 0.968 | | | |
| SCNEC | 43 | | | | | | | | | | | | |
| LCNEC | 2 | | | | | | | | | | | | |
| Age, years | | 1.458 | 0.538-3.944 | 0.459 | | | | 1.358 | 0.559-3.301 | 0.499 | | | |
| ≦45 | 21 | | | | | | | | | | | | |
| >45 | 24 | | | | | | | | | | | | |
| Tumor size(cm) | | 2.193 | 0.707-6.807 | 0.174 | | | | 1.339 | 0.534-3.361 | 0.534 | | | |
| <4 | 19 | | | | | | | | | | | | |
| ≧4 | 26 | | | | | | | | | | | | |
| FIGO stage(2018) | | 2.623 | 1.094-6.288 | 0.031 | 0.882 | 0.309-2.516 | 0.815 | 2.673 | 1.221-5.852 | 0.014 | 0.716 | 0.129-3.964 | 0.702 |
| I | 8 | | | | | | | | | | | | |
| II | 28 | | | | | | | | | | | | |
| III | 9 | | | | | | | | | | | | |
| Histological homology | | 1.435 | 0.534-3.851 | 0.474 | | | | 1.821 | 0.750-4.420 | 0.185 | | | |
| Pure | 24 | | | | | | | | | | | | |
| Mixed | 21 | | | | | | | | | | | | |
| Pelvic LN metastasis | | 2.165 | 0.744-6.299 | 0.156 | | | | 2.653 | 1.049-6.713 | 0.039 | 1.706 | 0.622-4.683 | 0.300 |
| No | 35 | | | | | | | | | | | | |
| Yes | 10 | | | | | | | | | | | | |
| Lymphovascular invasion | | 4.207 | 1.441-12.285 | 0.009 | 3.241 | 1.105-9.505 | 0.032 | 3.055 | 1.211-7-708 | 0.018 | 1.562 | 0.556-4.392 | 0.398 |
| Negative | 25 | | | | | | | | | | | | |
| Positive | 20 | | | | | | | | | | | | |
| Depth of stromal invasion | | 4.203 | 1.173-15.056 | 0.027 | 2.36 | 0.598-9.320 | 0.220 | 3.520 | 1.152-10.754 | 0.027 | 2.639 | 0.846-8.230 | 0.095 |
| Inner third | 17 | | | | | | | | | | | | |
| Middle to outer third | 28 | | | | | | | | | | | | |
| Neoadjuvant therapy | | 1.171 | 0.403-3.399 | 0.771 | | | | 1.032 | 0.394-2.700 | 0.949 | | | |
| No | 15 | | | | | | | | | | | | |
| Yes | 30 | | | | | | | | | | | | |
| Chemotherapy regimen | | 0.782 | 0.352-1.740 | 0.547 | | | | 0.929 | 0.473-1.825 | 0.831 | | | |
| TP | 17 | | | | | | | | | | | | |
| EP | 20 | | | | | | | | | | | | |
| Other regimens | 6 | | | | | | | | | | | | |

Chen et al. (2023), *PeerJ*, DOI 10.7717/peerj.15084

Chen et al. (2023), *PeerJ*, DOI 10.7717/peerj.15084

**Table 5** (*continued*)

| Variable | n | Overall survival | | | | | | Progression free survival | | | | | |
|---|---|---|---|---|---|---|---|---|---|---|---|---|---|
| | | Univariate | | | Multivariate | | | Univariate | | | Multivariate | | |
| | | HR | 95%CI | *P* | HR | 95%CI | *P* | HR | 95%CI | *P* | HR | 95%CI | *P* |
| Cycle of chemotherapy | | 0.48 | 0.173-1.334 | 0.159 | | | | 0.747 | 0.293-1.903 | 0.54 | | | |
| 0-5 | 21 | | | | | | | | | | | | |
| $\geqq$6 | 24 | | | | | | | | | | | | |
| $ADC_{min}(10^{-3}mm^2/s)$ | | 11.49 | 1.515-87.163 | 0.018 | 1.440 | 0.096-21.552 | 0.792 | 6.983 | 1.616-30.173 | 0.009 | 2.084 | 0.287-15.12 | 0.468 |
| <0.680 | 28 | | | | | | | | | | | | |
| $\geqq$0.680 | 17 | | | | | | | | | | | | |
| $ADC_{max}(10^{-3}mm^2/s)$ | | 16.69 | 2.230-128.977 | 0.006 | 14.413 | 1.883-110.342 | 0.01 | 6.667 | 1.945-22.850 | 0.003 | 5.668 | 1.634-19.66 | 0.006 |
| $\leqq$1.032 | 24 | | | | | | | | | | | | |
| >1.032 | 21 | | | | | | | | | | | | |
| $ADC_{mean}(10-3mm^2/s)$ | | 4.194 | 1.447-12.160 | 0.008 | 1.257 | 0.345-4.578 | 0.729 | 2.740 | 1.118-6.717 | 0.028 | 3.655 | 0.639-20.91 | 0.145 |
| $\leqq$0.700 | 18 | | | | | | | | | | | | |
| >0.700 | 27 | | | | | | | | | | | | |

**Notes.**

SCNEC, Small cell neuroendocrine carcinoma; LCNEC, large cell neuroendocrine carcinoma Adjuvant therapy includes chemotherapy, radiotherapy and concurrent chemoradiation; CT, chemotherapy; CCRT, concurrent chemoradiation; NACT, neoadjuvant chemotherapy; EP, etoposide and cisplatin/carboplatin; TP, paclitaxel and cisplatin/carboplatin..

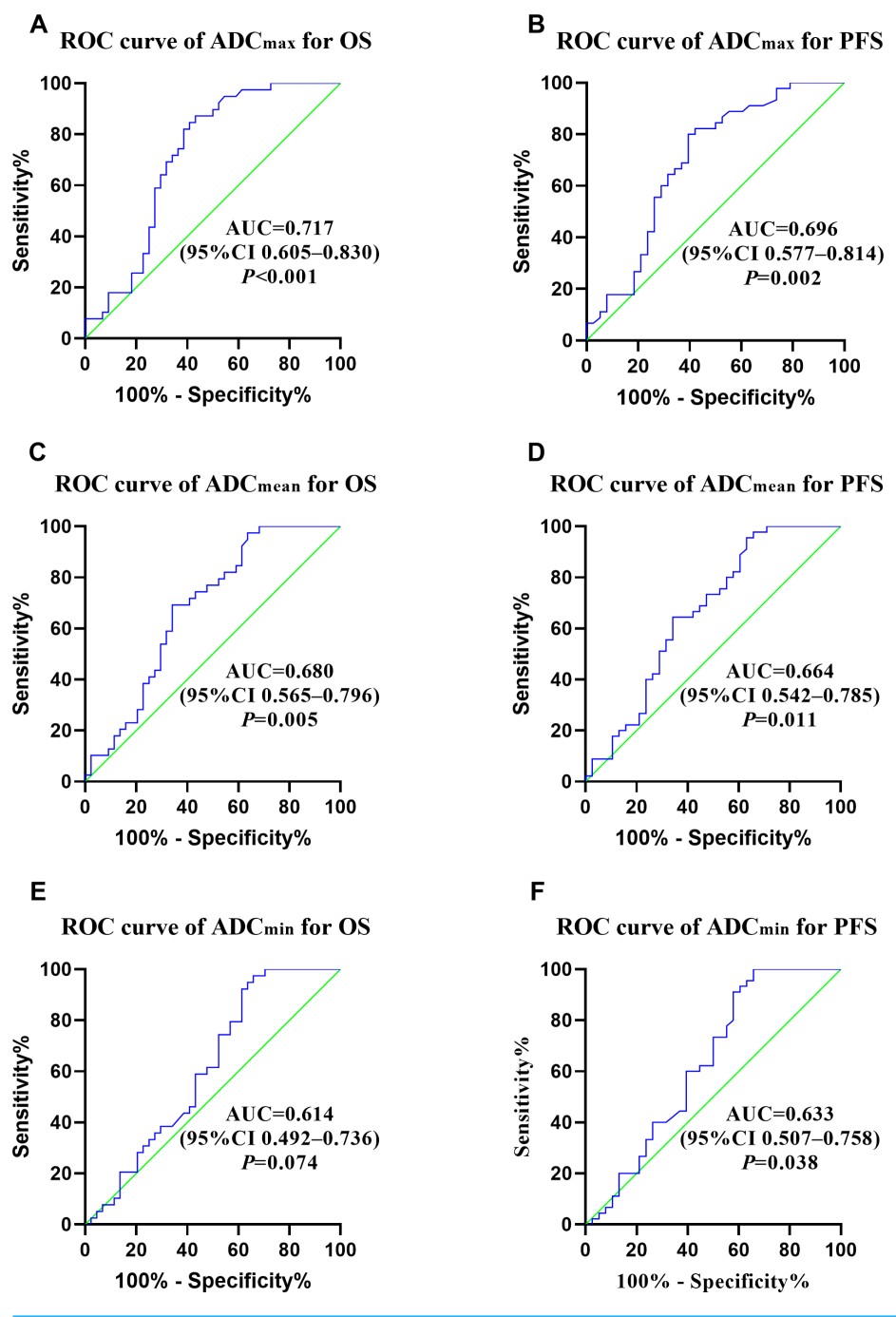

**Figure 2** (A–F) ROC curves of ADC$_{max}$, ADC$_{mean}$, ADC$_{min}$ for OS and PFS.

has been reported for the first time. We also assessed the accuracy of MRI in the diagnosis of NECCs. It was previously suggested that the decreased ADC values in malignant tumours indicated proliferative activity and increased tissue cellularity, which led to the disordered arrangement of the intracellular structure and decreased extracellular spaces (*Nakamura et al., 2012*; *Schob et al., 2017b*). Additionally, some studies have reported that ADC values

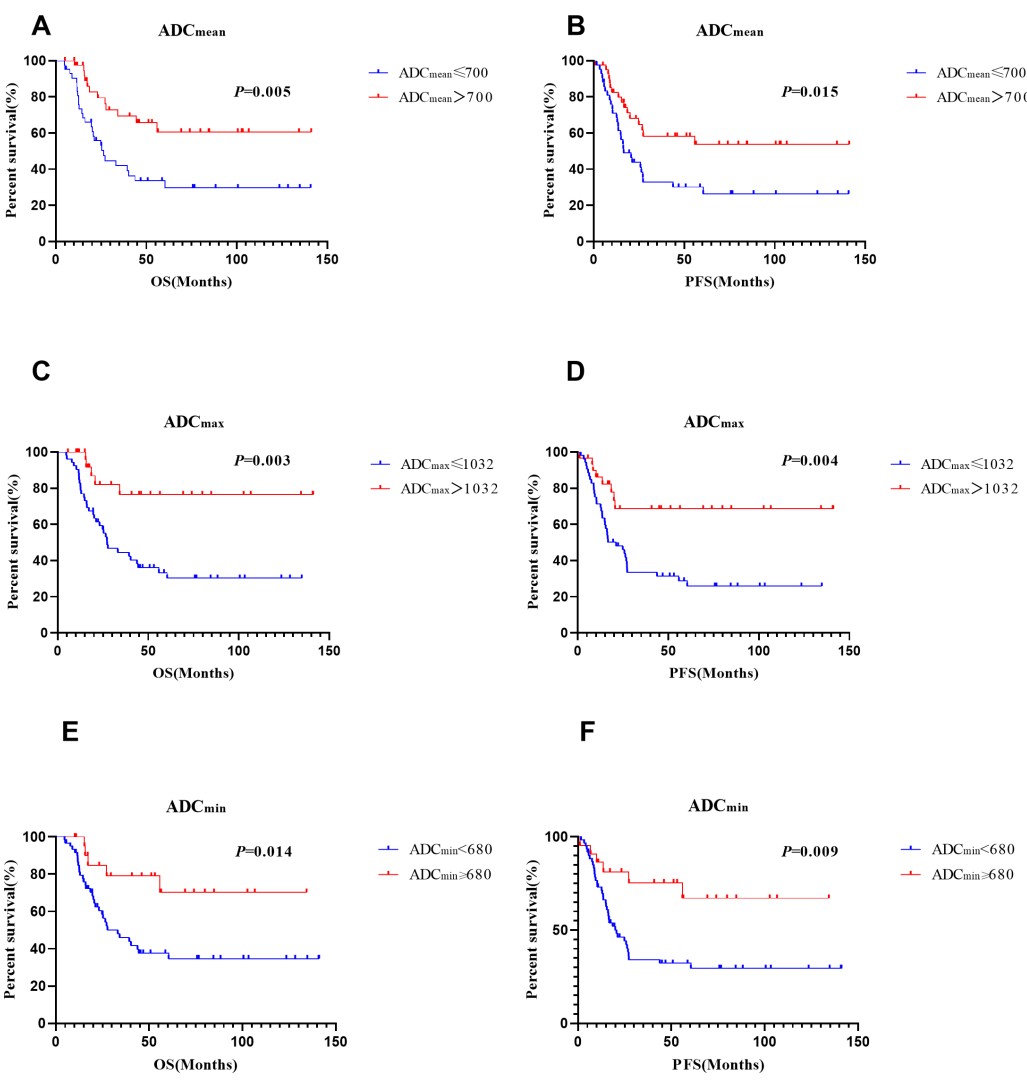

**Figure 3** **Survival curves of different ADC vaules.** (A), (C) and (E) for OS, (B), (D) and (F) for PFS.

reflect the tumour aggressiveness and predicted prognosis and treatment response to chemoradiation therapy. *De Robertis et al. (2018)* found that ADC maps may help predict the tumour grade, vascular involvement, and nodal and liver metastases in pancreatic neuroendocrine tumours. *Heo et al. (2013)* showed that pre-treatment ADC values could predict the tumour recurrence in patients who were diagnosed with cervical cancer and treated with chemoradiation. They found that patients with the 75th percentile ADC >0.936 $\times 10^{-3}$ mm$^2$/s had significantly better overall recurrence free survival raterates than those with the 75th percentile ADC ≤0.936 $\times 10^{-3}$ mm$^2$/s (91.7% *vs.* 51.9%, $p = 0.003$) . Other researches have also shown that ADC analysis may be an effective clinical biomarker to forecast treatment response and survival rate among patients with hepatocellular carcinoma and rectal cancer (*Choi et al., 2016*; *Shaghaghi et al., 2020*). However, the efficacy of ADC values in predicting FIGO staging and prognosis in patients with NECCs is unclear.

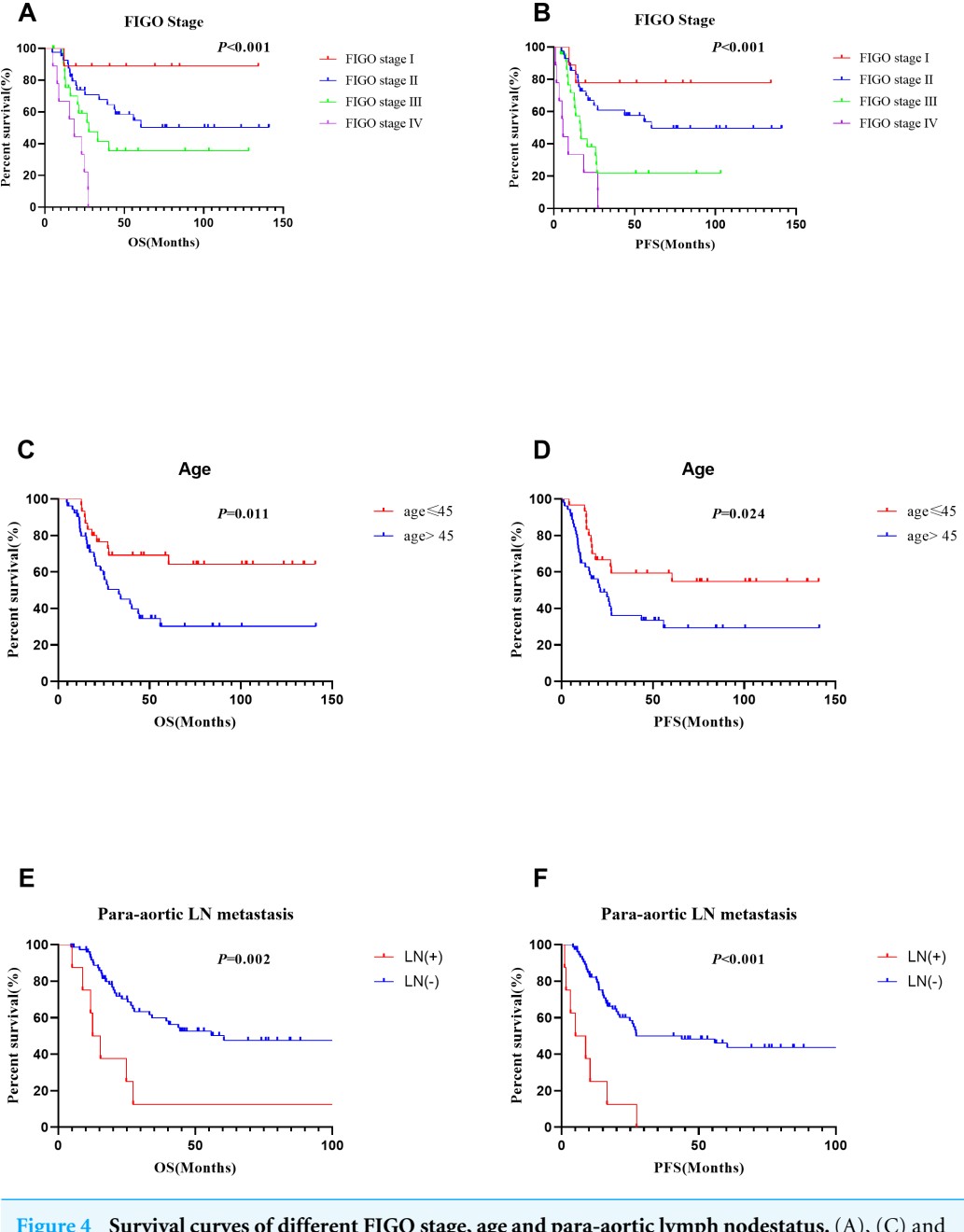

**Figure 4** **Survival curves of different FIGO stage, age and para-aortic lymph nodestatus.** (A), (C) and (E) for OS, (B), (D) and (F) for PFS.

Therefore, we explored whether pre-treatment ADC values were associated with clinicopathological characteristics in patients with NECC in this study. It was observed that lower ADC values were greatly associated with advanced FIGO stage, large tumour size, deep stromal invasion, and pelvic lymph node metastasis. We used the pre-treatment MRI and demonstrated that ADC values were greatly related to the prognosis in patients with NECCs. In multivariate analysis, $ADC_{mean} \leq 0.7 \times 10^{-3}$ mm$^2$/s were associated with

wores overall survival rates (HR =2.344, $p = 0.018$) and ADC $_{min}$ $\leq 0.68 \times 10^{-3}$ mm$^2$/s were associated with worse progression-free survival rates (HR =3.787, $p = 0.006$). These findings are similar to those of a prior study. Zhao et al. reported that pre-treatment ADC$_{min}$ was significantly correlated with the disease-free survival in patients with cervical cancer (HR = 0.110, $p = 0.006$) (*Zhao et al., 2019*). In surgically treated patients, we found that ADC $_{max}$ $\leq 1.032 \times 10^{-3}$ mm$^2$/s was greatly associated with worse OS and PFS. The risk of recurrence and disease progression increased by 14.4 and 5.7 times, respectively, compared with those with ADC $_{max}$>$1.032 \times 10^{-3}$ mm$^2$/s. ROC curve analyses were performed to decide which ADC value among these three values performing the best ability. ADC $_{max}$ seems to have better diagnostic effectiveness than ADC$_{mean}$ and ADC$_{min}$, and the AUC of ADC$_{max}$ for OS and PFS was 0.717 and 0.696, respectively.

This research showed that FIGO stage and age at diagnosis were prognostic factors for OS, and FIGO stage, age at diagnosis, and para-aortic lymph node metastasis were prognostic parameters for PFS; these results are similar to those of our previous study (*Chen et al., 2021*). Additionally, the lymphovascular invasion was founded to be a prognostic parameter for 5-year OS in the patients who underwent surgery; however, FIGO stage was not. This may be due to the fact that the proportion of lymphovascular invasion in patients with stage I disease (50.0%) was higher than that in patients with stage II disease (32.1%).

MRI is regarded as a useful and accurate method for the diagnosis of cervical tumours. *Sala et al. (2007)* found that MRI was 83% (8/41) accurate in diagnosing cervical carcinomas with vaginal invasion. They assumed that this inaccuracy was caused by a large exophytic cervical tumour stretching the vaginal fornix. A meta-analysis of 57 studies showed that the sensitivity for parametrial invasion in MRI was 74% (*Bipat et al., 2003*). In our study, the accuracy rates of MRI in the diagnosis of uterine corpus invasion, parametrial invasion, vaginal invasion, and lymph node metastasis were 86.7%, 80.0%, 53.3%, and 93.3%, respectively. There were two false-positive parametrial invasions. Cervical biopsy or cervical conisation was performed before MRI examination, resulting in inflammation and stromal oedema; this resulted in an inadequate estimation of parametrial invasion. Stromal oedema caused by tumour compression is also a possible cause (*Nakamura et al., 2012*; *Park et al., 2014*). Additionally, *Woo et al. (2019)* found that for determining parametrial invasion, oblique axial T2WI may be more accurate than true axial T2WI, especially for tumours larger than 2.5 cm.

According to a meta-analysis of 72 studies comprising 5,042 patients, MRI exhibited a sensitivity of 56% and a specificity of 93% for detecting lymphadenopathy (*Choi et al., 2010*). For detecting lymph node metastasis, the size criterion used in MRI was a short axis diameter $\geq$ one cm (*Balleyguier et al., 2011*; *Dappa et al., 2017*). This criterion, however, is flawed because it overlaps with normal, hyperplastic, and metastatic lymph nodes. Furthermore, micrometastases in negative lymph nodes are not rare (*Lee, Kim & Park, 2020*). Therefore, we considered round shape, irregular border, and necrosis as other signs of malignancy (*Balleyguier et al., 2011*). In our study, the sensitivity, specificity and accuracy of MRI in detecting lymph node metastasis were 50.0%, 100%, and 93.3%, respectively. *Lin et al. (2008)* observed that the method combining tumour size and ADC values had better sensitivity (25% *vs.* 83%) and similar specificity (98% *vs.* 99%) compared

 

with those of the traditional MRI approach. MRI was useful in detecting the parametrial invasion, uterine corpus invasion, and lymph node metastasis; however, the diagnostic efficacy of vaginal invasion needs to be improved.

There are several limitations in this research. First, this was a retrospective study; therefore, selection bias was unavoidable. Second, we did not measure ADC values for the entire tumour in this study. Further studies using histogram analyses will be needed. Moreover, ROIs were drawn manually by two radiologists, and measurement errors were inevitable. Third, the study did not include dynamic contrast-enhanced MRI, which is a useful diagnostic tool. Fourth, para-aortic lymph node metastasis affected OS in our research as an independent prognostic parameter; however; only 22% (10/45) of the patients had para-aortic lymph node dissection. Finally, this was a single-centre study with a small sample size, especially for those with LCNECs. More studies involving larger cohorts are needed.

## CONCLUSION

It was observed that lower ADC values were greatly associated with advanced FIGO stage, large tumour size, deep stromal invasion, and lymph node metastasis. $ADC_{mean}$ and $ADC_{min}$ were independent prognostic parameters for NECCs. For surgically treated patients ($n = 45$), $ADC_{max}$ was an independent prognostic parameter for both 5-year OS and PFS. Additionally, we found that MRI is reliable for the prediction of uterine corpus invasion, parametrial invasion, and lymph node metastasis, but not vaginal invasion. ADC analysis may be a useful tool for predicting the FIGO stage and outcome of patients with NECCs.

## ACKNOWLEDGEMENTS

We thank Xiaojie Wang for the helpful suggestions and comments on the manuscript.

### Funding

This work was supported by the Startup Fund for scientific research, Fujian Medical University (Grant number: 2020QH1216). The funders had no role in study design, data collection and analysis, decision to publish, or preparation of the manuscript.

### Grant Disclosures

The following grant information was disclosed by the authors:
Fujian Medical University: 2020QH1216.

### Competing Interests

The authors declare there are no competing interests.

### Author Contributions

- Jian Chen conceived and designed the experiments, performed the experiments, prepared figures and/or tables, authored or reviewed drafts of the article, and approved the final draft.

- Ning Ma conceived and designed the experiments, performed the experiments, prepared figures and/or tables, authored or reviewed drafts of the article, and approved the final draft.
- Mingyao Sun performed the experiments, prepared figures and/or tables, authored or reviewed drafts of the article, and approved the final draft.
- Li Chen performed the experiments, prepared figures and/or tables, authored or reviewed drafts of the article, and approved the final draft.
- Qimin Yao performed the experiments, prepared figures and/or tables, and approved the final draft.
- XingFa Chen analyzed the data, authored or reviewed drafts of the article, and approved the final draft.
- Cuibo Lin performed the experiments, authored or reviewed drafts of the article, and approved the final draft.
- Yongwei Lu analyzed the data, authored or reviewed drafts of the article, and approved the final draft.
- Yingtao Lin analyzed the data, authored or reviewed drafts of the article, and approved the final draft.
- Liang Lin analyzed the data, authored or reviewed drafts of the article, and approved the final draft.
- Xuexiong Fan analyzed the data, authored or reviewed drafts of the article, and approved the final draft.
- Yiyu Chen performed the experiments, authored or reviewed drafts of the article, and approved the final draft.
- Jingjing Wu performed the experiments, authored or reviewed drafts of the article, and approved the final draft.
- Haixin He conceived and designed the experiments, performed the experiments, analyzed the data, prepared figures and/or tables, and approved the final draft.

## Data Availability

The raw measurements are available in the Supplementary File.

## Supplemental Information

Supplemental information for this article can be found online at http://dx.doi.org/10.7717/peerj.15084#supplemental-information.

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
