# Peer review of "Prognostic value of apparent diffusion coefficient in neuroendocrine carcinomas of the uterine cervix"

_PeerJ, doi:10.7717/peerj.15084_

## Round 0.1 · original submission · Minor Revisions

Thank you for submitting your manuscript to PeerJ, which has been through the peer-review process. Reviewer comments are below. When revising your manuscript, please carefully consider all issues mentioned in the reviewers' comments: please outline every change made in response to their comments and provide suitable rebuttals for any comments not addressed. Please note that your revised submission may need to be re-reviewed.

Reviewer 1 ·

Basic reporting

no comment

Experimental design

This is a well-written article. The author explored the prognostic value of ADC values in predicting FIGO stage and outcome of patients with neuroendocrine carcinomas of the uterine cervix (NECCs). As the neuroendocrine carcinomas is rare kind of histological type, the sample size of the study is not small relatively.
In the imaging analysis part of Materials & Methods, line 115, the author wrote the sentence as “The ADC map was constructed automatically using the Functool software. Regions of interest (ROIs) were drawn along the edge of the enhancing region of the lesions showing maximal tumour size manually.” The “enhancing region” may confuse readers that the ROIs were made by ADC map or contrast-enhanced images or DWI images. Please clarify it.

Validity of the findings

In the Conclusion, the author said “ADCmean, ADCmin, and ADCmax are independent prognostic factors for NECCs”, I am interested in which ADC value among these three values performing the best ability. The difference among the three ADC values need to discuss more for readers.

Additional comments

In result part of Abstract, in the third line, the word " dianosis" spelling is not right. In the discussion, line 227, the word“cclinicopathological”is a spelling error.

Reviewer 2 ·

Basic reporting

This paper presents the results of the ADC values as an independent prognostic factor for NECCs, and ADC analysis could be useful in predicting the survival outcomes in patients with NECCs. The topic discussed in the paper is interesting and highly relevant, given that NECCs is an infrequent but highly invasive form of cervical cancer. I think the findings reported in the paper provide added value for the scientific community and can provide advice for clinical practice.

Experimental design

1. The patient cohort is not properly described. The patients with histologically confirmed NECCs were selected, but only 15 patients were confirmed with pathological results. Thus, a more detailed histologically diagnose processing should be provided.

2. Some methods and preview works lack references.
a) Page 7 “The ADC map was constructed automatically using the Functool software.” should cite the website of SPSS.
b) Page 8 “The SPSS version 24.0 statistical package (SPSS Inc.)” should cite the website of SPSS.
c) In the discussion section, “The current research showed that FIGO stage and age at diagnosis were prognostic factors for OS, and FIGO stage,” should add the reference of the described research.

Validity of the findings

1. Figure titles and annotations were not given in the review vision.

2. Figure 2 contained black arrows which confused me, and the digits up the black arrows are also incomprehensible, for instance, the “1.041” in figure 2A. According to my guess, the digits might be the cutoff values for OS and PFS. In my opinion, including those digits might be unnecessary.

Additional comments

Multiple spelling and grammar errors should be corrected.

a) In the Image analysis section, “the the tumour size”.
b) In the introduction section, “Advance FIGO stage, lymph node involvement”.
c) In the Prognostic factors section, “Figures 3 and Figures 4 show survival curves”.

---

## Round 0.2 · accepted · Accept

The authors successfully addressed the majority of concerns raised by the reviewers. Therefore I recommend the article for publication.

Reviewer 1 ·

Basic reporting

No comment

Experimental design

No comment

Validity of the findings

No comment